# Health Related Quality of Life Measurements for Diabetes: A Systematic Review

**DOI:** 10.3390/ijerph18179245

**Published:** 2021-09-01

**Authors:** Sampson Emilia Oluchi, Rosliza Abdul Manaf, Suriani Ismail, Hayati Kadir Shahar, Aidalina Mahmud, Theophilus Kachidelu Udeani

**Affiliations:** 1Department of Community Health, Faculty of Medicine and Health Sciences Universiti Putra Malaysia, Serdang 43400, Selangor Darul Ehsan, Malaysia; oluzube@yahoo.com (S.E.O.); si_suriani@upm.edu.my (S.I.); hayatik@upm.edu.my (H.K.S.); aidalina@upm.edu.my (A.M.); 2Department of Medical Laboratory Sciences, Faculty of Health Sciences & Technology College of Medicine, University of Nigeria Enugu Campus, Enugu 410001, Nigeria; theophilus.udeani@unn.edu.ng

**Keywords:** diabetes, quality of life, HRQOL, instruments, measurement

## Abstract

Health-related quality of life (HRQOL) is an essential measure that is used to assess the effect of chronic disease management on the health status of an individual. Previous studies have identified various instruments used in the measuring of diabetes-specific health-related quality of life (HRQOL). The aim of this paper is to provide a systematic review of the various instruments used for the diabetes-specific measure of HRQOL, and place emphasis on its content and measurement properties. Methods Preferred Reporting Items for Systematic Reviews and Meta analyses (PRISMA) guidelines was used. A systematic search strategy was used to identify publications reporting diabetes HRQOL measures. The search terms used were: “diabetes quality of life”, “measurements”, and “instruments”. The database that was searched includes PubMed, Science Direct, CINAHL, and Medline. Articles written in the English language and published from January 1990 to December 2020 were included. Those articles that did not measure HRQOL for diabetic patients were excluded. Results: A total of seventeen instruments met the inclusion criteria and included in the review. The appraisal of diabetes scale (ADS), Audit of Diabetes-Dependent QOL measure (ADDQOL), Diabetes Health Profile (DHP), and Problem Areas in Diabetes (PAID) are more suitable for single-scale questionnaires when investigating one or more specific aspects of diabetes-specific quality of life (QOL). The ADDQOL, ADS, Diabetes Impact Measurement Scales (DIMS), Diabetes Quality of Life Clinical Trial Questionnaire (DQLCTQ-R), Malay Version of Diabetes Quality of Life (DQOL), Iranian Diabetes Quality of Life (IRDQOL), Brief Clinical Inventory, and PAID are relevant measures of HRQOL for insulin dependent diabetes mellitus (IDDM) and non-insulin dependent diabetes mellitus (NIDDM) patients. The Asian Diabetes Quality of Life AsianDQOL, The Chinese Short Version of DQOL, Elderly Diabetes Burden Scale (EDBS), Malay Version of Diabetes Quality of Life (DQOL), are relevant measures of HRQOL for NIDDM patients. Only two instruments assess for responsiveness, namely PAID and DQLCTQ-R. In PAID, the effect sizes ranged from 0.32 to 0.65 for interventions. The DQLCTQ-R four domains were responsive to clinical change in metabolic control. Based on this review ADDQOL, DSQOLS, and EDBS psychometric properties are sufficient. Conclusion: Most studies did not check for responsiveness, and future studies should prioritize responsiveness to change, which was not included in the psychometric finding of the reviewed instruments.

## 1. Introduction

Quality of life (QOL) describes overall satisfaction with life, either as a single concept or broken down into domains [1]. It is how an individual feels about his or herself, whether their life be good or bad. QOL represents illness outcome of the patient, as perceived by the patient, and produces information to medical or epidemiological data, which are used as measurement outcomes. QOL is known as a multidimensional concept comprising domains regarding the general well-being, the future physical health and functioning, mental health, satisfaction with treatment, and social functioning [2]. In QOL research, subjective QOL deals with an individual’s good feeling and how generally satisfied they are with things, and not how others imagine them to be. Objective QOL measure is about how fulfilled the societal and cultural material wealth demands are, including social status and physical well-being, in both a research setting and clinical practice [3]. Nevertheless, QOL is a central issue for patients, providers, and policymakers. Recently, interest in health-related quality of life (HRQOL) has increased. The word ‘Health-related Quality of Life’ is used in the sense that it includes aspects of life which are not usually considered as ‘health’ [4]

The effects of diabetes on HRQOL are defined in literature. The illness and its complications, treatments, and the attitudes of patients work together to impair HRQOL’s multiple dimensions, which comprises of physical, social, cognitive, role, emotional well-being, sexual functioning, pain, and health perceptions. Health-related quality of life (HRQOL) is important in examining people’s health outcomes. The HRQOL relates to the degree of people’s life and their overall well-being, as well as their life satisfaction levels as either good or bad [5]. HRQOL is important because it is used to assess the consequence and management effectiveness of health condition problems or chronic diseases [6]. Chronic illness research outcomes are progressively becoming concerned with evaluations of patients’ clinical effectiveness of treatment and care. Relevant health outcomes include both physiological measures and subjective factors such as self-management of disease burden, emotional health and physical functioning, and social and role functioning [7]. For people with diabetes mellitus, these subjective factors are significant because the disease is mainly self-managed, and self-management affects almost all daily life aspects.

The problem of diabetes and its influence on HRQOL is a public health concern to patients, families, employers, healthcare workers, and taxpayers [8]. Diabetes disease, its complications, treatments, and subsequent patient attitudes damage HRQOL dimensions, which include social, physical, emotional well-being, cognitive and sexual functioning, pain, and health perceptions [9]. Diabetes requires continual adjustments in health behaviors and a ‘compliance to strict prescribed treatment’ [10,11,12]. Various instruments have been developed, validated, and used to measure the HRQOL of diabetic patients. There are substantial numbers of HRQOL measurements precise to diabetes. Thus, this can be unclear for researchers and clinicians that have an interest in measuring the HRQOL of patients with diabetes but are faced with numerous instruments with diverse measurement approaches. The aim of this paper is to provide a systematic review of the various instruments used for the diabetes-specific measuring of HRQOL, and to place emphasis on their content and measurement properties.

## 2. Materials and Methods

Preferred Reporting Items for Systematic Reviews and Meta-analyses (PRISMA) guidelines were used [13]. A systematic search strategy was used to detect publications reporting diabetes HRQOL measure for diabetes patients (Figure 1). The search terms used were: “diabetes”, “quality of life”, “diabetes quality of life”, “survey”, “measurements”, and “instruments”. The database searched included PubMed, Science Direct, CINAHL, PsycINFO, and Medline. Titles were checked to exclude unrelated articles. Abstracts were read and duplicated studies were removed. Retained studies were reviewed in full text. Articles retrieved were screened for eligibility criteria. The articles were included for full-text article reviews in the case there is doubt regarding the abstracts. Independently, the two investigators reviewed all full-text articles to confirm if inclusion criteria were met or not. All articles retrieved during the search was assessed independently by two authors of the team. Each article title and abstract were reviewed by two authors.

## 3. Eligibility Criteria

Inclusion criteria in this review were restricted to instruments with a primary focus on the development, reliability, and validity of disease-specific HRQOL measures. Selected studies were measurements developed for insulin dependent diabetes mellitus (IDDM) patients and non-insulin dependent diabetes mellitus (NIDDM) patients, in addition to original research and full-text articles written and published in the English language from January 1990 to December 2020. Lastly only the instrument that could be accessed was included. Exclusion criteria included articles that did not measure HRQOL for diabetes. Studies that focused on children and adolescents were also excluded because they were descriptive reports and did not include methodology and measurement issues. Studies that focused on gestation diabetes were excluded. This systematic review addresses instruments evaluated in adult diabetic patients.

## 4. Results

A total of 602 studies published from January 1990 to December 2020 were identified through electronic searching using search key words. The 20 articles identified through other sources were identified from Google and Google Scholar. The titles were read, and the duplicated and unrelated articles were removed. After removing duplicates and articles not related to the study, it yield 420 studies. A total of 420 studies were related to be included, and the abstracts were read. Among the 420 articles, only 41 articles were included to be reviewed in full text. Finally, inclusion and exclusion criteria were applied. Among the 41 articles, only 17 articles were selected and included in the review. The 24 articles that were excluded were not original articles; they were review papers. Among the 17 articles included in the review, eight studies involved both IDDM patients and NIDDM patients, eight articles involved only NIDDM, and one article involved only IDDM patients. Three studies were conducted in Malaysia [14,15,16]; seven studies were conducted in the United States of America [17,18,19,20,21,22,23]. One study was conducted in China [24], two studies were conducted in the United Kingdom [25,26], and one study was conducted in Iran [27]. One study was a multinational one, and one study was conducted in Germany [28]. One study was conducted in Japan [29]. Table 1 shows the list of selected publications and the different types of HRQOL measures used. Moreover, Table 2 shows the psychometric information of the instruments.

## 5. Various Quality of Life Measurements for Diabetes

There are various approaches to assess HRQOL among diabetic patients. Various instruments have been used to measure HRQOL among diabetic patients, which leads to difficulty when choosing instruments for research. The instruments are classified as generic or specific. Generic HRQOL measures make available useful information regarding the health status of diabetic patients and compares them with other populations and other groups of chronic diseases. The disease-specific HRQOL measures focus on dimensions unique to diabetes. Thus, examples are worries of diabetes symptoms, self-care, treatment satisfaction, and adherence to diabetic regimen, social and family support. The instruments have been used to measure HRQOL in IDDM and NIDDM patients, and some of them are included in this review.

The Diabetes Quality of Life (DQOL) instrument was used in diabetes research for years to measure the quality of life for diabetic patients. Although the DQOL was developed for the evaluation of IDDM, it has also been used for NIDDM [19,30]. The DQOL instrument has 46 items and was used to measure HRQOL among diabetic patients. It was ranked on a 5-point Likert scale, and is based on four main domains, namely, “satisfaction”, “impact”, diabetes related worry, and social/vocational worry. The DQOL also includes 16 items that measures the education and relationships of young people with their family [31]. The DQOL was revised for the Diabetes Control and Complications Trial (DCCT) [19,32].

The DQOL has a strong reliability and has been confirmed to be valid [32,33]. The internal consistency of DQOL was reported as fair, with a Cronbach alpha coefficient of 0.47 to 0.87 [19]. The validity was based on content validity. Content validity was carried out among experts [32].

The validity and stability of DQOL have been proven, and the DQOL instrument is commonly used for diabetes research [34,35]. The limitation of this instrument is that it has numerous items which represent the three main domains. However, it is essential to cover various viewpoints of quality of life among patients with diabetes. There are other well-known quality of life instruments that contain large items such as the Diabetes Obstacles Questionnaire (DOQ), which is comprised of 113 items [26], and the Diabetes Quality of Life Clinical Trial Questionnaire (DQLCTQ), which is comprised of 142 items [36]. Questionnaires with many items needs more time to complete. Respondents may fill up irrelevant answer because of insufficient time to reason before responding. This may lead to large number of missing values, and therefore cause frustration to the researcher because items were not answered appropriately.

The Diabetes Quality of Life Revised Version (DQOL-R), was recently published in 2018. The intention of the DQOL revised version was to develop a shorter form of the DQOL instrument; it maintained the assessment of the three main domains “satisfaction”, “impact”, and “worry”. The DQOL revised version instrument was concluded with 13 items; the “satisfaction” domain has six items, the impact domain has four items, and the “worry” domain has three items. DQOL revised version was introduced to back-up validity with quantitative measures using exploratory factor analysis, confirmatory factor analysis, and Rasch analysis. Reliability of every domain was also computed: “satisfaction” domain was seen with the highest reliability of 0.922, the “worry” domain was 0.794, and was 0.781 for “impact” domain [14].

Nevertheless, the DQOL revised version is unable to validate the original DQOL instrument; no study was able to validate DQOL successfully using exploratory factor analysis. This is because some of the items are not appropriate for majority of people with diabetes. One example is the question regarding sexual life, which is sensitive for some countries [37]. In addition, this item is not appropriate for patients who are not sexually active. The main strength of the revised version of the DQOL instrument is that it has less items (only 13), and, hence, less time is needed to complete the questionnaire. Thus, the revised version of the DQOL instrument has a benefit, since a study revealed that questionnaires with many items are less likely to be completed [38]. Besides being a shorter version, it is more appropriate among diabetes patients to reflect quality of life.

DQOL Brief Clinical Inventory is an instrument used to measure HRQOL among diabetic patients. This measurement was developed and validated to measure the diabetes-related QOL of diabetic patients [17]. It has 15 items and four domains relevant to perceptions of treatment, impact of treatment, treatment satisfaction, worries of future diabetes effects, and worry regarding issues of social/vocational. The DQOL instrument was developed for IDDM and NIDDM as part of the DCCT. The instrument has good internal consistency (r = 0.78–0.92), test–retest reliability (r = 0.78–0.92), and convergent validity for all four subscales for individuals with IDDM and NIDDM [19,32]. DQOL Brief Clinical Inventory makes available a total HRQOL score that predicts self-reported diabetes care behaviors and whether patients are satisfied with diabetes control as efficiently as they were with the full version of the instrument. It provides quick screening vehicle for the patients’ readiness and specific treatment-related concerns. Regarding the strength of the instrument, it can be used to detect quality of life problems that might not arise during the patient’s visit to their healthcare provider. The instrument takes around 10 min to administer [17].

The Malay Version of Diabetes Quality of Life (DQOL) was validated among adult patients with NIDDM in Malaysia [16]. The Malay version of diabetes quality of life (DQOL) instrument is a valid and reliable instrument used for adults with diabetes in Malaysia. The DQOL instrument was confirmed by these findings: a degree of correlation was found among the three major domains—a positive correlation was found between Impact Domain and HbA1c, and an association between diabetic complications and Worry Domain. The Cronbach’s alpha coefficients and of the three major domains were between 0.846 and 0.941. It was found that HbA1c was positively correlated with the Impact Domain (*p* = 0.003). The study findings revealed that the Malay version of DQOL questionnaire has an excellent internal consistency and validity for Malaysian adult diabetic population. Regarding the strength of the instrument, it was that, based on the face validity of the instrument, most patients could understand the questionnaire easily. In addition, the Malay version of the DQOL questionnaire reported an excellent internal consistency level within each domain.

The Chinese Short Version of DQOL was based on two psychometric theories, the classical test theory (CTT) and the item response theory (IRT), each combined with the exploratory factor analysis (EFA), respectively. It has 24-items and four domains, namely time satisfaction to manage diabetes, duration of time to receive a checkup, “the time to determine sugar level”, and “present treatment”. The Chinese short version of DQOL used a 5-point Likert scale. The CFA and Spearman correlation coefficients were used to validate the two short versions. The short version of the Chinese DQOL was provided by CTT with 32 items and the short version with 24 items was provided by IRT. The CTT excluded 14 items and the IRT excluded 13 items [25]. The Cronbach’s alpha for the two factors exceed 0.7 (Factor 1: 0.884 Factor 2: 0.822). A larger corrected item–total correlation coefficient indicated better internal consistency reliability. Concerning strengths and weaknesses, the Chinese short version of DQOL was selected as the preferred short version, this was because it enforces a lower burden on patients without compromising the instrument’s psychometric properties. The instrument does not have a truly external validation sample. The training sample only contained community-based patients. The instrument cannot be generalized to the whole population of diabetic patients, as the sample was relatively healthier than the diabetic population who had more comorbidities, was inpatient, or using insulin.

Iranian Diabetes Quality of Life (IRDQOL) is comprised of 41 items, including one specific to married patients and one specific to unmarried patients among adults with IDDM and NIDDM diabetes [28]. The questions from the 41 items were qualitative research, covering general and HRQOL. The instrument’s Cronbach’s alpha coefficient was 0.98. The correlation coefficient of the IRDQOL instrument was 0.639 and it was acceptable to the concurrent criterion validity. The score was 40–160, higher scores demonstrates improved QOL. Assessing general QOL 13 items was used concerning fatigue, loneliness, calmness, worry, tension, spirituality, and financial issues. Items measuring HRQOL were based on the physical and psychosocial conditions of the effect of diabetes. For general QOL scores, 13–52 and 27–108 for HRQOL. Better QOL in both cases is indicated by higher scores.

Asian Diabetes Quality of Life (AsianDQOL) was validated in an English, Malay, and Chinese–Mandarin pilot study [15]. For comparison, the World Health Organization Brief Quality of Life Questionnaire (WHOQOL-BREF) was used. For the English language, a focus group of 30 subjects with NIDDM was used, including ten Malaysian, ten Chinese, and ten Indian participants. For the AsianDQOL (English), 74 points or below was considered as a poor score, a moderate score was 75–81, a good and above was 82–88, and an excellent score was 88 points for QOL. Regarding the Malay language, five components of 21 items were demonstrated for EFA. The AsianDQOL (Malay) poor QOL was considered as 76 points or below, moderate was 77–85, good and above was 86–91, and excellent QOL was 91 points—this is nearly similar to the English AsianDQOL system of scoring. The EFA for the Chinese–Mandarin version had 18 items and 5 components. The AsianDQOL Chinese (Mandarin) scoring was not consistent with the median. Scores of less than 65 points were considered poor, moderate was 65 to 70, good and above was 71 to 79, and excellent QOL was 80 points. The AsianDQOL total score is unique to the individual language. The total score can also be used to classify the global quality of life score for patients. However, this scoring system was based on a cross-sectional study and a small sample.

The Cronbach’s alpha scores (English version) was 0.917, the Malay version was 0.833, the Chinese/Mandarin version 0.890. The AsianDQOL is a valid, reliable, and stable tool for assessing QOL in multiethnic and multilingual T2DM Asian populations.

The AsianDQOL is more appropriate for the Malaysian population compared to DQOL, DQLCTQ-R, and DSQOLS because it is disease-specific and was created based on the Malaysian population [15].

The Diabetes Quality of Life Clinical Trial Questionnaire (DQLCTQ-R) was developed based on DQLCTQ. The DQLCTQ-R comprises of 57 questions and eight generic and also disease-specific domains: frequency of symptoms, physical function, energy/fatigue, health distress, mental health, satisfaction, treatment flexibility, and treatment satisfaction [39]. The draft developers of the questionnaire added previously validated measures (SF-36 and DQOL) and new items were developed as desired. Intraclass correlation coefficients range from 0.74 to 0.90 and Cronbach’s alpha ranges from 0.77 to 0.90. The items are Likert-scaled and it takes 10 min to complete the questionnaire. A strength of the instrument is that the DQLCTQ-R is a valid, reliable, and comprehensive HRQOL instrument. It is also appropriate to use in multinational clinical trials to evaluate new or alternative treatments for IDDM and NIDDM patients. The instrument was translated to French and German.

The Diabetes Obstacles Questionnaire (DOQ) was developed and validated to measure obstacles for people living with NIDDM [26]. It contains eight subscales; the instrument is valid for both in research and clinical purposes. The DOQ originally contained 113 items. It was developed based on an academic literature review. The items are based on a 5-point scale labeled strongly agree, neutral, disagree, and strongly disagree. Eight statements were connected to when diabetes was diagnosed (i.e., in what way individuals were told the first time that they have diabetes and their feelings when they were diagnosed), 13 statements were linked knowledge of diabetes, 17 statements were linked to the medical treatment of diabetes, 16 statements linked to communication between the healthcare provider and patient, and also 59 statements were linked to the patients’ adherence to the diabetes regimen (i.e., lifestyle change, problems with blood glucose self-monitoring [26]. It has a Cronbach alpha coefficient of 0.75. A strength of DOQ is that it covers a wider and more detailed range of problems and obstacles than in the Problem Areas in Diabetes (PAID) method [40]. It recognizes in detail the difficulties with people living with NIDDM.

The Audit of Diabetes-Dependent QOL measure (ADDQOL) is an instrument that is designed to measure an individual’s perceptions of the impact of diabetes on their quality of life. The 13 specific domains included in the version of the ADDQOL [18], which are employment chances, family relationships, friendships, sex life, social life, sporting, holiday, the ease with which I can travel, worries regarding household future and close friends, physical things to do, and the extent people would worry about me. The items were scored on a seven-point scale and the respondent determined whether the item is very important, important, quite important, or not important at all. NIDDM and personal invitations were issued primarily for people with IDDM and was advertised in the local press. The coefficient in excess of 0.8 suggests that, for some purposes, the scale might usefully be shortened. The items do not differ markedly in their effects on the reliability data and, in this instance, provide a useful basis for excluding items. The instrument ADDQOL reported the item–total correlation, which is ranged from 0.28 to 0.84 [19]. The ADDQL scoring ignores non-applicable domains and more emphasis is given to the domains’ individual rates, which are deemed as more important. Diabetes had more influence on diabetes-specific domains, for example, pleasure of food, worries regarding upcoming events and travel, than on standard QOL domains, for example, work, social life, family, and friends, suggesting that diabetes-specific ADDQOL is extra sensitive to change [41]. Globally, ADDQOL is the most translated and validated questionnaire.

The Diabetes-specific Quality of life Scale (DSQOLS) has 64 items, and the scale was developed based on a review of current diabetes-specific QOL questionnaires and group discussions with IDDM patients, and it was reviewed by diabetes healthcare experts [28]. The scale was developed to measure the QOL of IDDM patients. The scale has six domains: social relations, leisure tile flexibility, physical complaints, worries regarding the future, diet restrictions, and daily hassles. It was based on a six-point Likert scaled. The Cronbach’s alpha for the six domains exceeded 0.7. Cronbach’s alpha is recommended to exceed 0.7 [41]. To complete the questionnaire, it takes less than 20 min. In a previous study, five subscales were included to the questionnaire [42]. It was translated from German to English for use in the UK. The DSQOLS is specific for IDDM patients [28].

The Diabetes Care Profile (DCP) instrument was developed to measure psychological and social factors associated with diabetes and its treatment [20]. The questionnaire has 234 items and it is comprehensive. It was derived from the educational profile of diabetes and the health belief model. To some extent, it deals with matters related to beliefs, knowledge of diabetes, and treatment. However, six subscales of the DCP measure diabetes-specific QOL domains comprising of personal, social, emotional functioning, and perceptions of control. It takes 30 to 40 min to complete the questionnaire. The DCP is unique because it is comprehensive and covers the social and phonological aspects and treatment. DCP has three scales, which were significantly correlated with glycated haemoglobin level. Using the DCP scale highlights worsening QOL and is associated with higher glycaemic levels and the use of insulin or tablets if the patient is having a larger number of complications due to diabetes [20,43,44]. Ethnicity had no influence on the DCP scoring scale [45]. DCP did not report the internal consistency reliability of the Cronbach alpha coefficient.

The Diabetes Health Profile (DHP) was developed to examine psychological well-being related to diabetes and focuses on psychological aspects. The Diabetes Impact Measurement Scales (DIMS) has 40 items. The instrument was developed based on a review of literature, previous instruments, and discussions with diabetes healthcare experts. The scale was designed to measure changes in longitudinal health status among diabetic patients, and the instrument was used for clinical trials [21]. The Items were grouped into four subscales: general well-being, physical symptoms, social functioning, and diabetes-related morale. Items were based on four and six-point Likert scales. Cronbach alpha test was carried out for the internal consistency of the subscales and the total DIMS scale: the values obtains were within the range considered desirable by psychometric standards. A strength of the instrument is that the questionnaire is simple and straightforward; comprising of items that are easily understood, it covers a broad range of content relevant to diabetes’ impact. It takes 15 to 20 min to complete this questionnaire. The instrument was translated into Chinese, French, and Italian, including the domains of distress, barriers to activity, and dietary perceptions and behavior [26]. The content of the instrument was obtained based on a literature review, a review of existing instruments, interviews with diabetic patients, and discussions with diabetes healthcare experts. Validated original studies resulted in a 32-item, three factor questionnaire, developed for use among insulin-requiring and insulin-dependent diabetic patients above 18 years old. The DHP reliability coefficients for internal consistency (Cronbach’s standardized alpha) for psychological distress were cr = 0.85, for barriers to activity a = 0.85, and for disinhibited eating a = 0.80, all of which were satisfactory. Psychological distress and barriers to activity were the two most correlated scales. The Cronbach alpha coefficient for the three domains exceeded 0.7. The reliability coefficients for internal consistency (Cronbach’s standardized alpha) were cr = 0.85 for psychological distress, a = 0.85 for barriers to activity, and a = 0.80 for disinhibited eating. The instrument’s DHP reported the item–total correlation, ranging from 0.28 to 0.84 [22]. Hyperglycemic complaint of fatigue had a significantly negative impact on psychological distress and barriers to activity [12]. The instrument was revised in a cross-cultural study to a shorter version with 18 items with the same three factors, and was modified for use among NIDDM patients (DHP-18).

The Appraisal of Diabetes Scale (ADS) is comprised of seven items based on theory and previous research [23]. It was developed to assess an individual’s appraisal of his or her diabetes. The seven items use a five-point scale and measure control, uncertainty, coping, the effect of diabetes on life goals, predictive view of diabetes and the degree of distress caused by diabetes. The ADS has been used to assess the effects of family environment and work environment on glycemic control and of psychosocial adaptation among diabetic adults [45,46]. The internal consistency of ADS was reported as fair, with a Cronbach alpha coefficient of 0.7. It takes less than five minutes to complete the questionnaire.

A strength of ADS is that the ADS is easy to score and interpret, which can be administered by nonprofessional support staff. The questionnaire can be completed within five minutes. Moreover, the questionnaire can be easily read for patients who are illiterate or have a visual impairment. The instrument has face validity and it inquires only diabetic-related information. The ADS could prove useful as a brief screening instrument for diabetes adjustment. The instrument can be administered to diabetic patients to identify patients that are experiencing or at risk of dysphoric reactions and noncompliance issues.

The Elderly Diabetes Burden Scale (EDBS) is a short version of the elderly diabetes impact scale (EDIS). EDBS is a measure of diabetic-specific QOL for diabetes mellitus elderly patients; the scale has 37 items [30]. Among the 37 items, 23 selected items were rated on a four-point multiple-choice scale, from which the authors developed the EDBS [47]. The EDBS has six subscales which include worry about diabetes, symptom burden, treatment dissatisfaction, burden by tablets or insulin, dietary restrictions and social burden. The EDBS subscales and total EDBS have no significant correlations with the adaptive feeling to diabetes and the examination of mini-mental state; this shows a discriminant validity of EDBS [48]. It takes less than five minutes to complete the questionnaire. It is translated from Japanese to English.

The Problem Areas in Diabetes Scale (PAID) is a 20 item scale, the scale is used to measure diabetes-related distress and it was developed by researchers associated with the Joslin Diabetes Center and Harvard Medical School [22]. The 20-item PAID scale is based on emotional problems. Items were developed based on patient interviews, input from diabetes healthcare experts, and also a pilot study. Items were rated on a six-point Likert scale. The questionnaire takes less than five minutes to complete. Based on the PAID scores, there was no significant difference between subjects with IDDM and NIDDM, as age (r = −0.11) and disease duration (r = −0.12) were weakly correlated. The Cronbach alpha coefficient was 0.95, which indicates a high level of internal reliability. The PAID could be a clinically valuable instrument in measuring psychosocial adjustment to diabetes. The PAID correlated positively with levels of HbA1c, and it scales were moderately to strongly associated with related measures of general and diabetes-specific stress [22].

The PAID score was also related to psychosocial distress, disordered eating, fear of hypoglycemia, and complications. No differences in the scores were found between INIDDM and NIDDM. The PAID is a brief and easy to administer instrument and may serve as a clinical tool useful in the identification of patients who are experiencing high levels of diabetes-related distress. Globally, PAID is among the most translated and validated questionnaire.

## 6. Discussion

There are numerous studies on HRQOL, which has led to the development of new instruments and modifications of some diabetes-specific HRQOL. This systematic review focuses on specific instruments developed for measuring HRQOL for diabetic patients.

Such instrument have been used in diabetes research for years to measure quality of life for diabetic patients. The DQOL was developed for the evaluation of IDDM, it has also been used for NIDDM [19,30] and has numerous items, and a questionnaire with many items needs more time to complete it. Respondents may fill up irrelevant answers because of insufficient time to reason before responding. This may lead to large number of missing values. The DQOL revised version was able to develop a shorter form of the DQOL instrument; however, the DQOL revised version is unable to validate the original DQOL instrument, therefore no DQOL study was able to be validated successfully using exploratory factor. The Malay version of the DQOL questionnaire reported an excellent internal consistency level within each domain. The Chinese short version of DQOL instrument cannot be generalized to the whole population of diabetic patients, as the sample was relatively healthier than the diabetic population who had more comorbidity, were inpatient, or using insulin.

The AsianDQOL is more appropriate for the Malaysian population compared to DQOL, DQOL-R, DSQOLS, and DQLCTQ-R because it was created based on the Malaysian population [15].

ADDQOL and ADS place emphasis on the stressful impact of life with diabetes, respectively. Brief Clinical Inventory provides a quick screening vehicle for patients’ readiness and specific treatment-related concerns. The following instruments—DQOL, DSQOLS, DCP, and DIMS—are more suitable for a study with a broad conceptualization of diabetes, specifically QOL. The DQLCTQ-R was developed for use in multinational clinical trials, and it has pertinent domains related to HRQOL. DHP is concentrated on diabetes-related distress, activity, and eating manners; the PAID is concentrated on diabetes-related distress. EDBS is concentrated on diabetic-specific QOL among elderly diabetic patients.

All of the reviewed instruments were developed in the English language. Among the reviewed instruments, the instrument that has less items takes less time to complete the questionnaire.

Studies have revealed that questionnaires with many items are less likely to be completed [39]. Based on the reviewed instruments, in one or more respects, validation is lacking. The DCP, DIMS, and ADS did not involve patients in the construction of its items. Patients should be involved in the derivation of items, especially when the instrument is to have content validity as a measure applicable to the recipients of care [48].

The MalayDQOL, AsianDQOL, DOQ, and DHP are specific to NIDDM. The DSQOLS is specific to IDDM. The other reviewed instruments are appropriate measures of HRQOL for IDDM and NIDDM patients.

The reviewed instruments produce acceptable reliability estimates, apart from DQOL and DIMS, which have weaker evidence. The reviewed instruments reported internal consistency, as measured by Cronbach’s alpha coefficient.

Among the reviewed instruments, only DQLCTQ-R and PAID assessed for responsiveness to changes in health, the authors of the DSQOLS mentioned an evaluation of a teaching programme among IDDM patients as evidence for responsiveness [29]. However, one major inadequacy of patient assessed measures among diabetes patients is the lack of testing for responsiveness to changes in health. Further studies are needed to evaluate the responsiveness based on longitudinal comparisons of measurements within clinical trials. One of the reviewed studies validated, EDBS, focused only among the elderly populations. Further studies should be conducted among adults with diabetes.

A systematic review has been published on HRQOL previously among diabetic adult patients; however, this current review is an update. In the previous diabetes-specific HRQQL review, nine instruments were reviewed. According to the authors, among the nine instruments, five of them proved internal and external validity and also reliability [49]. Another systematic review was carried out, and twelve diabetes-specific HRQQL instruments were identified. The authors pointed out that the twelve instruments reviewed presented conceptual and measurement methods and stated that applying them in diabetes research will be useful for understanding the construct of HRQOL [50]. Finally, another systematic review on diabetes-specific HRQQL was carried out, and sixteen instruments were reviewed. Their study focused on the description of the measurements and the findings [51].

Based on our review, the psychometric findings of the instruments shows that there is variation in the content of the seventeen diabetes-specific instruments reviewed. It is important for potential users when selecting diabetes-specific QOL instruments to consider the content of the instruments in relation to the patient population and research question.

Numerous measures exist for the assessment of HRQOL in diabetic patients or for assessing the disruption of diabetes, suggesting that, in order to avoid the unnecessary development of new instruments, it would be more suitable to choose an already existing and validated instrument.

Regarding recommendations for future study, diabetes research offers a broad array of conceptual approaches, it is therefore recommended to continually use diabetes-specific instruments of HRQOL as, this will further the understanding of the important constructs. Moreover, conceptual underpinnings of the numerous instruments are to be considered, such as the use of diabetes-specific instruments of HRQOL measures to empower patients and physicians relationships.

An analysis of the strengths and weaknesses of the seventeen diabetes-specific instruments reviewed shows that HRQOL is operationalized within multidimensional surveys comprising of personal, social functioning, emotional functioning, life satisfaction, treatment satisfaction, perceptions of control, mental, physical, and social health components. It is important to keep the context and the purpose for which the specific tool was developed in mind: Was it intended for clinical use, or instead for research? Is it patient-centered, treatment-centered, or diet-centered? Clinically oriented measure could be used for research and vice-versa, different purposes generally require different characteristics.

## 7. Conclusions

Diabetes has great impact on the QOL of diabetic patients. Regarding the instruments of disease-specific HRQOL in diabetes, DQOL, DQOL-R, DQOL Brief Clinical Inventory, ADDQOL, DHP, DSQOLS, and EDBS have good psychometric properties, as the authors involved patients when developing these instruments. Although most studies did not check for responsiveness, future studies should prioritize responsiveness to change, which was not included in the psychometric finding of the reviewed instruments. Concurrently, instruments that are to be used to enable relative responsiveness and convergent validity need to be evaluated. There is need to examine the effects of ethnicity and to determine the validity of these scales in developing countries in further research.

## Figures and Tables

**Figure 1 ijerph-18-09245-f001:**
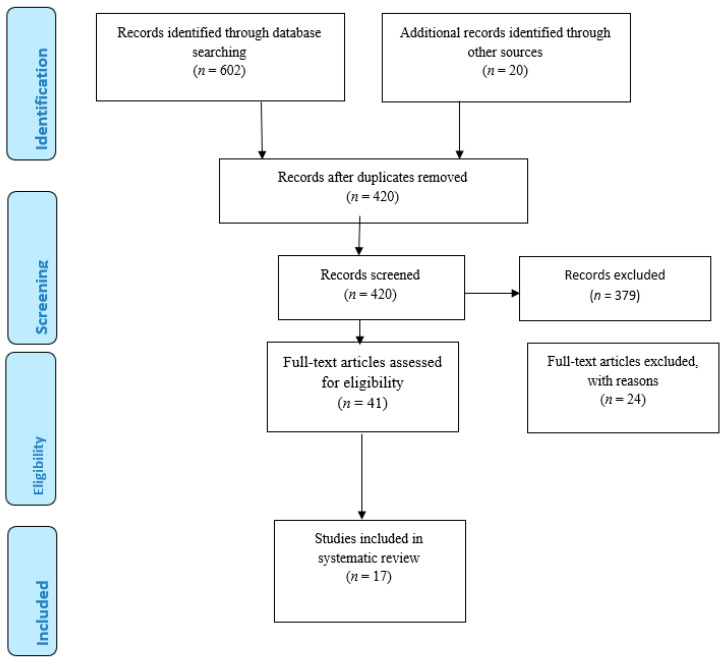
PRISMA flow Diagram.

**Table 1 ijerph-18-09245-t001:** List of selected publications and the different types of HRQOL measures used.

Title/Author/Year of Publication	Country	Name of Instrument Used	Domains of HRQOL Used	Strength and Weakness
The development of an individualized questionnaire measure of perceived impact of diabetes on quality of life: The ADDQOL. Bradley et al. (1999)	United Kingdom	Audit of Diabetes-Dependent QOL measure (ADDQOL)	It has 13 Domains: Employment/Career Opportunities, Social Life, Family Relationships, Friendships, Sex, Life, Sporting, Holiday or Leisure Opportunities; The Ease with which I can Travel; Worries about my Future; Worries about the Future of my Family and Close Friends; Motivation to Achieve Things; Things I could do Physically and the Extent to which People would Fuss too much about Me	Diabetes-specific ADDQOL will be more sensitive to change and responsive to subgroup differences than a generic instrument such as the SF-36.
Reliability and validity of the appraisal of diabetes scale, Carey et al. (1991)	United States of America	Appraisal of diabetes scale (ADS)	Not mentioned	The ADS is easy to score and interpret and can be administered by nonprofessional support staff. The questionnaire can be completed within five minutes. The ADS could prove useful as a brief screening instrument for diabetes adjustment. The instrument can be administered to diabetic patients to identify patients that are experiencing or at risk of dysphoric reactions and noncompliance issues.
Development and validation of the Asian Diabetes Quality of Life Questionnaire Goh, Rusli, and Khalid (2015).	Malaysia	Asian Diabetes Quality of Life (AsianDQOL)	Not mentioned	The total score for AsianDQOL is unique to the respective language. To review the instruments used to assess the impact of PPC interventions. The AsianDQOL is more suitable for use in Malaysian population compared to DQOL, DQLCTQ-R and DSQOLS because it is disease specific and was constructed based on the Malaysian. The AsianDQOL is a valid, reliable, and stable tool for assessing QOL in multiethnic and multi-lingual NIDDM Asian populations
Item reduction and validation of the Chinese version of diabetes quality-of-life measure, Jin et al. (2018).	China	Chinese short versions DQOL	Four domains: Satisfaction level of “the amount of time it takes to manage your diabetes”; “the amount of time you spend getting a checkup”; “the time it takes to determine your sugar level”; “your current treatment”	Chinese DQOL was the preferred short version because it imposes a lower burden on patients without compromising the psychometric properties of the instrument. Training sample contained community-based patients, and most of them were not using insulin. This sample was relatively healthier than the diabetic population, who had more comorbidities, was inpatient, or using insulin; thus, the results cannot necessarily be generalized to the entire diabetic patient population.
Development and Validation of the Diabetes Care Fitzgerald et al. (1996).	United States of America	Diabetes Care Profile (DCP)	The instrument comprises of six subscales of the DCP measure diabetes-specific QOL domains comprising of Personal, Social, Emotional Functioning, and Perceptions of Control.	Using the DCP scale, results of worse QOL are associated with higher glycaemic levels, use of insulin or tablets, and if the patient is having larger number of complications due to diabetes. It takes 30 to 40 min to be complete the questionnaire.
The diabetes health profile (DHP): a new instrument for assessing the psychosocial profile of insulin requiring patients: development and psychometric evaluation, Meadows et al. (1996)	United Kingdom	Diabetes Health Profile (DHP)	Three subscales: The three factors were interpreted as Psychological Distress, Barriers to Activity and Disinhibited Eating.	DHP appears to be a reliable and valid instrument suitable for further development and application in a clinical and research context.
Measurement of Health Status in diabetic patients: Diabetes Impact Measurement Scales, Diabet. Care. Hammond and Aoki (1992)	United States of America	Diabetes Impact Measurement Scales (DIMS)	The items were grouped into four subscales: General Well-Being, Physical Symptoms, Social Functioning, and Diabetes-related Morale.	The Diabetes Impact Management Scales is an easily administered questionnaire with internal consistency and test–retest reliability. The questionnaire is simple and straightforward, comprising of items that are easily to understood; it covers a broad range of content relevant to diabetes impact.
Development and validation of the Diabetes Obstacles Questionnaire (DOQ) to assess obstacles in living with Type 2 diabetes Hearnshaw et al. in (2007).	United Kingdom	Diabetes Obstacles Questionnaire (DOQ)	DOQ, comprising of eight subscales covering Medication, Self-Monitoring, Knowledge and Beliefs, Diagnosis, Relationships with Health-Care Professionals, Lifestyle Changes, Coping, and Advice and Support.	DOQ covers a much wider and more detailed range of problems and obstacles than the Problem Areas in Diabetes (PAID). DOQ is a usable and valid instrument for both clinical and research settings. It helps to identify in detail the obstacles which an individual finds in living with NIDDM.
Development and validation of the diabetes quality of life clinical trial questionnaire Shen et al. (1999)	Multinational study: United States of America, Canada, Germany, and France	Diabetes Quality of Life Clinical Trial Questionnaire (DQLCTQ-R)	Energy/Fatigue; Health Distress; Mental Health; Satisfaction; Treatment Satisfaction; Treatment Flexibility; Frequency of Symptoms.	It is appropriate to use for IDDM and NIDDM patients.
Reliability and validity of a diabetes quality-of-life measure for the diabetes control and complications trial Jacobson, Barofsky, Cleary, and Rand, (1988).	United States of America	Diabetes Quality of Life (DQOL)	Four Domains: Life Satisfaction, Diabetes Impact, Social/Vocational Related Worries, and Diabetes-related Worries.	The Diabetes Control and Complications Trial (DQOL) questionnaire has 46 items developed for IDDM diabetes as part of the DQOL. It is particularly relevant for the worry scales, because they were developed especially for use in younger patient samples. DQOL in its full form is too lengthy to be completed as part of a provider’s routine office visit.
A revised version of diabetes quality of life instrument maintaining domains for satisfaction, impact, and worry. Bujang et al. (2018)	Malaysia	Diabetes Quality of Life Revised version DQOL-R	“satisfaction” domain has six items, impact domain has four items, and “worry” domain has three items.	It has lesser items, only 13 items, and, hence, less time is needed to complete the questionnaire.
Validation of a diabetes-specific quality-of-life scale for patients with type 1 diabetes, Bott et al. (1998)	Germany	Diabetes-specific Quality of life Scale (DSQOLS)	The scale has six domains: Social Relations, Leisure Tile Flexibility, Physical Complaints, Worries Regarding the Future, Diet Restrictions and Daily Hassles.	To complete the questionnaire takes less than 20 min.
Development and Validation of the Diabetes Quality of Life Brief Clinical Inventory. Burroughs et al. (2004)	United States of America	DQOL Brief Clinical Inventory	Four domains: Satisfaction with Treatment, Impact of Treatment, Worry about the Future Effects of Diabetes, and Worry about Social/Vocational Issues.	The 15-item DQOL Brief Clinical Inventory provides a total health-related quality of life score that predicts self-reported diabetes care behaviors and satisfaction with diabetes control as effectively as the full version of the instrument. In addition, it provides a vehicle for quickly screening patients for readiness and specific treatment-related concerns. It takes about 10 min to administer and can be used to identify quality of life issues that might not arise during the typical patient provider encounter.
Development of elderly diabetes impact scales (EDIS) in elderly patients with diabetes mellitus, Araki et al. (1995)	Japan	Elderly Diabetes Burden Scale (EDBS)	The EDBS has six subscales which include Worry about Diabetes, Symptom Burden, Treatment Dissatisfaction, Burden by Tablets or Insulin, Dietary Restrictions, and Social Burden	The EDBS is useful in evaluating the quality of life in elderly patients with diabetes mellitus.
Developing a culturally valid and reliable quality of life questionnaire for diabetes mellitus. Alavi, Ghofranipour, Ahmadi, and Emami, (2007).	Iran	Iranian Diabetes Quality of Life (IRDQOL)	Not mentioned	The questionnaire successfully distinguished the lower QOL of patients suffering from pain in the limbs, loss of appetite, fatigue, constipation, and itching. The questionnaire could determine both general and health-related QOL for IDDM patients.
Validation of the Malay version of Diabetes Quality of Life (DQOL) Questionnaire for Adult Population with Type 2 Diabetes Mellitus., Bujang, et al. (2017)	Malaysia	Malay version of Diabetes Quality of Life (DQOL)	Three domains, namely Satisfaction Domain, Impact Domain, and Worry Domain.	The Malay version of diabetes quality of life (DQOL) questionnaire was found to be a valid and reliable survey instrument to be used for Malaysian adult patients with diabetes mellitus.
Assessment of diabetes-related distress, Polonsky et al. (1995)	United States of America	Problem Areas in Diabetes Scale (PAID)	Not mentioned	The PAID is a brief and easy to administer instrument, which may serve as a clinical tool useful in the identification of patients who are experiencing high levels of diabetes-related distress.

**Table 2 ijerph-18-09245-t002:** Psychometric evaluation of diabetes specific health related QOL measures.

Instrument	Reliability		Validity	Responsiveness
	Cronbach’s α	Test– Retest	Scale Analyses	
Audit of Diabetes-Dependent QOL measure (ADDQOL)	0.85–0.92	-	Factor analysis: All items loading >0.40 on one factor, all items loading >0.50 on one factor; item–total correlations: 0.37–0.67	-
			Differences between groups Better QOL associated with: non-insulin treated patients; less frequent hypoglycemia; fewer disease complications; flexible dietary regimen	
Appraisal of diabetes scale (ADS)	0.73	0.85–0.89	Principal components analysis: single factor explaining 39% variance; item–remainder correlations 0.28–0.59	-
			Convergent validity Diabetes Health Belief Questionnaire-Revised r = 0.31–0.42; Diabetic Daily Hassles Scale r = 0.59; Perceived Stress Scale r = 0.39–0.58	
Asian Diabetes Quality of Life (AsianDQOL)	0.719–0.917	0.60	Confirmatory factor analysis: GFI = 0.88	-
			Differences between groups The component of diet and eating habits were significant in both the English language and Chinese–Mandarin versions but were not in the Malay language	
Chinese short versions DQOL	0.884	-	Confirmatory factor analysis Standardized root mean squared residual 0.078, Comparative fit index 0.726	-
	0.822			
Diabetes Care Profile (DCP)	0.60–0.95	-	Confirmatory factor analysis: GFI = 0.92	-
			Convergent validity Social Provisions Scale: r = −0.34 to 0.32 CES-D: r = −0.53–0.48; Happiness and Satisfaction Scale: r = −0.27 to 0.32	
			Differences between groups Not using insulin was associated with less impact on personal/social life, fewer control problems, positive outlook; number of complications (among those taking insulin) was associated with more impact on social/personal life	
Diabetes Health Profile (DHP)		-	Factor analyses: 33–35%, 32%, 40–46% of total variance explained; scale inter correlations: 0.13–0.57; item correlations: 0.47–0.75; inter-item correlations: 0.30–0.70	-
	0.77–0.86		External validity Coefficient of congruence: sex, 0.92–0.93; age, 0.93–0.99; language, 0.98–0.99	
	0.72–0.79		Convergent validity Hospital Anxiety and Depression Scale, r = 0.28–0.62; SF-36 r = −0.21 to −0.68, 0.07–0.65 (DHP items reverse-coded)	
	0.70–0.88		Differences between groups Younger women were more likely to be affected with psychological distress and eating disturbance than men	
Diabetes Impact Measurement Scales (DIMS)	0.60–0.94	-	Scale intercorrelations: 0.49–0.97; principal components analysis: single factor accounting for 32% variance	-
			Convergent validity Patient-rated diabetes control r = 0.22–0.55; Clinician-rated diabetes control r = 0.24–0.35; patient-rated general wellness r = 0.27–0.47; clinician-rated general wellness r = 0.29–0.45	
Diabetes Obstacles Questionnaire (DOQ)	0.766	-	variance explained ≥ 55%,	-
	0.813			
	0.866			
	0.834			
	0.937		Correlation coefficient 0.86–0.271	
	0.851			
	0.776			
	0.880			
Diabetes Quality of Life Clinical Trial Questionnaire (DQLCTQ-R)	0.77–0.89	-	Differences between groups Perceived control of diabetes is associated with better QOL, among male IDDM patients	Four domains were responsive to clinical change in metabolic control
Diabetes Quality of Life (DQOL)	0.67–0.88,	0.78–0.92	Scale intercorrelations: r = 0.26–0.68, 0.47–0.87 Test–retest: 0.78–0.92 Convergent/discriminatory validity Symptom Checklist Global Severity Index r = 0.40–0.77; Affect Balance Scale r = −0.25 to −0.67; Psychosocial Adjustment to Illness Scale r = 0.06–0.81; SF-36 r = −0.003 to 0.59 Differences between groups Adult males reported less diabetes impact, fewer worries than adult females; number of complications associated with less satisfaction had a greater impact; taking insulin associated with less satisfaction and a greater impact; not taking insulin associated with worry	-
Diabetes Quality of Life Revised version DQOL-R	0.67–0.88	0.78–0.92	Scale intercorrelations: r = 0.26–0.68, 0.47–0.87	-
			Convergent/discriminatory validity Symptom Checklist Global Severity Index r = 0.40–0.77; Affect Balance Scale r = −0.25 to −0.67; Psychosocial Adjustment to Illness Scale	
			r = 0.06–0.81; SF-36 r = −0.003 to 0.59	
Diabetes-specific Quality of life Scale (DSQOLS)	0.70–0.93	-	Goodness of fit index = 0.98; scale intercorrelations r = 0.28–0.66 Convergent validity Positive well-being scale r = 0.35–0.53	-
			Differences between groups Age r = _0.23–0.01; social status r = _0.04–0.24; better QOL associated with greater flexibility of insulin treatment, fewer complications and use of rapid-acting insulin	
DQOL Brief Clinical Inventory	0.61–0.94	-	Five significant principal components that accounted for 9.23–15.35% of the total item variance each and 56.73% of the total item variance collectively.	-
			Convergent validity Treatment satisfaction, the six-item model r = 0.254–0.562,	
			Differences between groups For worry about diabetes-related events, or for females for diabetes impact, no differences between the two groups	
Elderly Diabetes Burden Scale (EDBS)	0.55–0.89	0.94–0.99	Six-factor solution explaining 69.4% of variance	-
			Convergent validity Philadelphia geriatric center morale scale r = _0.51; Geriatric depression scale r = 0.27–0.57	
			Differences between groups It was reported that higher scores were seen among women’s dietary restrictions, worry, and less satisfaction of treatment, also more adaptive feeling to diabetes when compared to men	
Iranian Diabetes Quality of Life (IRDQOL)	0.98	-	Concurrent validity 0.639	-
			Differences between groups Quality of life has been found to be higher in males than females [22,23,24]. It seems sex can be considered	
Malay version of Diabetes Quality of Life (DQOL)	0.846–0.941	-	Correlation coefficients for the three domains were between 0.228 and 0.451	-
			**Differences between groups** Retinopathy group had a sizeable effect (mean score of 2.0 compared to no retinopathy group versus 2.7 from retinopathy group)	
Problem Areas in Diabetes Scale (PAID)	0.93–0.95	r = 0.83	Large single factor explaining 50–52% of variance; item–total correlations: r = 0.32 to 0. 84; all >0.30	Effect sizes range from 0.32 to 0.65 for interventions
			**Convergent validity** Global Severity Index of Brief Symptom Inventory r = 0.63; ATT39 r = −0.22 to −0.81; Diabetes Coping Measure-avoidance r = 0.05–0.59; Diabetes Coping Measure-passive resignation r = −0.01 to 0.70; Diabetes Coping Measure-tackling spirit r = −0.13 to −0.82; Well- Being Questionnaire r = −0.50 to −0.53; Hypoglycaemia Fear Survey (Worry) r = 0.53–0.57; State Trait Anxiety Inventory r = 0.61	
			**Differences between groups** IDDM reported more diabetes-related distress than NIDDM patients	

## Data Availability

Data sharing not applicable.

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
