# Peer review of "Health Related Quality of Life Measurements for Diabetes: A Systematic Review"

_ijerph, 2021, doi:10.3390/ijerph18179245_

Round 1

Reviewer 1 Report

The study presents a systematic review of literature, which is highly relevant for a broad audience. Thus, as such, it merits publication. Furthermore, the review is professionally laid out in all respects and professionally summarized.

My only important suggestion is that the authors should consider as to whether synergetic evidence occurred from putting known evidence together. As it stands now, the paper presents merely a pool of knowledge. If such new evidence occurred, it should be carefully stressed in the conclusion. If not, of course, then it is fine as it stands.

Besides, only a minor suggestion regarding the English style, which is generally fluent and easy to understand. However, there are here and there some sentence constructions that are not in full accordance with English grammar. I encourage a careful proofread to catch these.

Author Response

Thank you 

Numerous measures for the assessment of HRQOL in diabetic patients or assessing the disruption of diabetes, suggesting that In order to avoid an unnecessary development of new instruments, it would be more suitable to choose an already existing and validated instrument. Please refer to page 17.

Thank you,

English grammar has been improved.

Reviewer 2 Report

line 96 - whether the research included works on gestational diabetes or diabetes in pregnancy. If not, please complete it in the exclusion criteria.

Conclusions obtained from the study are very important in the context of the quality of life of patients with diabetes, the authors also set the direction of future research, which is an extremely important conclusion obtained from the meta-analysis. It is very important to describe the strengths and weaknesses of the analysis, which should be completed at the end of the chapter with a discussion.

Author Response

Thank you,

Studies that focused on gestation diabetes were excluded.  Please refer to page 4

Thank you,

 Strengths and weaknesses of the analysis, based on the seventeen diabetes specific instruments reviewed, analysis shows that HRQOL is operationalized with multidimensional surveys comprising of personal, social functioning, emotional functioning, life satisfaction, treatment satisfaction, perceptions of control, mental, physical and social health components.  It is important to keep in mind the context and the purpose for which the specific tool was developed: was it intended for clinical use, or instead for research? Is it patient-centered, treatment-centered, or diet-centered? While a clinically oriented measure could be used for research and vice-versa, different purposes generally require different characteristics. Please refer to page 17